# Comparison of the Intraocular Pressure-Lowering Effect of Minimally Invasive Glaucoma Surgery (MIGS) iStent Inject W and Hydrus—The 12-Month Real-Life Data [note 1]

**DOI:** 10.3390/diagnostics15040493

**Published:** 2025-02-18

**Authors:** Cedric Weich, Julian Alexander Zimmermann, Jens Julian Storp, Ralph-Laurent Merté, Nicole Eter, Viktoria Constanze Brücher

**Affiliations:** Department of Ophthalmology, University of Muenster Medical Center, Domagkstraße 15, 48149 Muenster, Germany; julian.zimmermann@ukmuenster.de (J.A.Z.); jens.storp@ukmuenster.de (J.J.S.); ralph-laurent.merte@ukmuenster.de (R.-L.M.); nicole.eter@ukmuenster.de (N.E.); viktoria.bruecher@ukmuenster.de (V.C.B.)

**Keywords:** minimal invasive glaucoma surgery (MIGS), Hydrus Microstent, iStent inject W, real-world clinical study

## Abstract

**Background/Objectives:** To compare the efficacy and safety of Hydrus^®^ Microstent and iStent inject^®^ W implants, in combination with phacoemulsification, for lowering intraocular pressure (IOP) in patients with glaucoma in a real-world clinical setting. **Methods:** This retrospective, single-center study analyzed medical records of glaucoma patients who underwent either Hydrus^®^ Microstent or iStent inject^®^ W implantation combined with cataract surgery at the University Hospital Muenster, Germany. Key outcome measures included absolute and relative IOP reduction, reduction in topical antiglaucoma medication use, overall success rate, and complications. A total of 32 eyes were analyzed, 16 in each treatment group, with a 12-month follow-up. **Results:** Both groups demonstrated significant postoperative IOP reduction (Hydrus: *p* < 0.001; iStent inject^®^ W: *p* = 0.032). The Hydrus group achieved significantly greater relative IOP reduction compared to the iStent inject^®^ W group (*p* = 0.043). The Hydrus group also showed a significant reduction in daily antiglaucoma medication use (*p* = 0.002), whereas the iStent inject^®^ W group did not achieve statistical significance in this regard (*p* = 0.054). The overall success rate was higher in the Hydrus group (38%) than in the iStent inject^®^ W group (13%), though the difference was not statistically significant (*p* = 0.102). No device-related complications were observed in either group. **Conclusions:** The Hydrus^®^ Microstent demonstrated superior IOP reduction and a more significant reduction in the need for antiglaucoma medications compared to the iStent inject^®^ W in a real-world setting. These findings support the use of the Hydrus implant as an effective and safe micro-invasive glaucoma surgery (MIGS) option for patients with mild to moderate POAG. Further studies with larger patient populations and long-term follow-up are warranted to confirm the efficacy in patients with PEX and advanced glaucoma.

## 1. Introduction

Glaucoma is a leading cause of blindness worldwide [1], characterized by the progressive loss of retinal ganglion cells and their axons, which leads to optic nerve damage. A key factor contributing to this degeneration is elevated intraocular pressure (IOP), although the disease’s pathophysiology is multifaceted and varies among individuals [2,3]. Elevated IOP remains the primary modifiable risk factor, and treatments aim to lower it through medications, laser therapy, or surgical interventions. The two primary mechanisms targeted in treatment are reducing aqueous humor production at the ciliary body or enhancing its outflow [4].

Traditionally, trabeculectomy (TE), which diverts aqueous humor into the subconjunctival space, has been the gold standard for surgical IOP reduction. However, in recent years, micro-invasive glaucoma surgery (MIGS) has gained prominence, particularly for patients with mild to moderate glaucoma. These procedures offer shorter operative times and cause less tissue disruption compared to more invasive surgeries [5].

One well-established MIGS device is the iStent inject^®^ (Glaukos, Aliso Viejo, CA, USA), which is inserted into the trabecular meshwork to enhance aqueous outflow into Schlemm’s canal. This implant is typically placed in combination with cataract surgery, with two stents positioned a few clock hours apart using a preloaded injector [6]. The newer iStent inject^®^ W, designed for easier placement, has shown efficacy in lowering IOP in patients with mild to moderate glaucoma, especially when combined with cataract surgery. Evidence suggests that implanting two to three micro-bypass stents is effective in managing mild to moderate open-angle glaucoma [7].

The Hydrus Microstent (Alcon, Freiburg, Germany) is another MIGS device that has been on the market for several years. Constructed from nitinol, an alloy of nickel and titanium, the Hydrus is an 8 mm trabecular stent designed to enhance outflow by dilating and scaffolding a portion of Schlemm’s canal (see Figure 1). It has demonstrated promising results in lowering IOP in patients with primary open-angle glaucoma (POAG), with the stent typically inserted to cover approximately one-quarter of the canal [8,9].

Numerous studies have demonstrated the success of both the Hydrus^®^ Microstent and the iStent inject^®^ W in specific patient populations. There are also some studies regarding the direct comparison between iStent and Hydrus. Some suggest a better IOP Reduction using Hydrus implantation [10,11]; other studies suggest no benefit in Hydrus implantation but more complications [12]. However, it remains unclear which of these two implants, in combination with phacoemulsification, yields superior outcomes in real-world settings [11,13]. In this study, we retrospectively evaluate the real-world data from a heterogeneous group of patients to compare the effectiveness of these implants.

## 2. Materials and Methods

This retrospective, single-center study was conducted in accordance with ethical standards established by the local ethics committee of the Medical Association of Westphalia-Lippe and the University of Muenster, following the principles of the Declaration of Helsinki. The study was carried out at the Department of Ophthalmology, University Hospital Muenster, Muenster, Germany.

Patient records from the FIDUS electronic medical record system (Arztservice Wente GmbH, Darmstadt, Germany) were analyzed. The study included patients diagnosed with glaucoma who underwent either the nitinol Hydrus^®^ Microstent implantation combined with phacoemulsification or the iStent inject^®^ W implantation with phacoemulsification. The surgical indication and the surgical procedure were determined by the attending senior physician. The operation was performed by three different experienced and certified doctors. Inclusion criteria required patients to have documented IOP readings 12 months postoperatively, data on postoperative antiglaucoma medications, and information regarding any additional IOP-lowering procedures or ocular adverse events following MIGS implantation. Patients who had previously undergone IOP-lowering procedures (e.g., filtering or cyclodestructive methods) or vitrectomy were excluded. The eligible Hydrus group was matched with a randomly selected cohort of iStent inject^®^ W patients. In total, 32 eyes were included in the study, and all characteristics are shown in Table 1.

All patients underwent a standardized ophthalmic evaluation, which included a refractive assessment, anterior segment examination, fundoscopy, Goldmann applanation tonometry, and automated perimetry using the Humphrey Visual Field Analyzer II (HFA II, model 750, Carl Zeiss Meditec AG, Jena, Germany) with the 30-2 SITA fast program. Fundus imaging of the optic disc was performed using the VISUCAM system (Zeiss, Jena, Germany).

Postoperatively, patients were categorized into one of three outcome groups in accordance with EGS (European Glaucoma Society) guidelines. Complete success was defined as achieving an IOP below 22 mmHg and a 25% IOP reduction 12 months after surgery without the need for antiglaucoma medications [14]. Qualified success was defined as achieving the same IOP targets with the aid of topical glaucoma medications. These two groups were collectively classified as overall success. Failure was defined as not meeting these criteria. Secondary and tertiary outcome measures included the relative reduction in IOP and the reduction in the use of topical antiglaucoma medications. In addition, we measured the percentage of patients who achieved the minimal clinically important difference (MCID) in terms of “drop-free” and an IOP reduction of at least 2 mmHg. The rate of complications was determined in accordance with EGS guidelines [14].

### Statistical Analysis

Data were extracted from the electronic medical records and analyzed using IBM SPSS Statistics software, version 29.0.0. For group comparisons with non-normal distribution, the Mann–Whitney U test was applied to independent groups, while the Wilcoxon matched-pairs signed-rank test was used for related samples. We used Pearson’s chi-squared test for the comparison of success rate. Statistical significance was defined as a *p*-value of less than 0.05.

## 3. Results

A total of 32 eyes were included in the study, with 16 eyes in each treatment group. There were no significant differences between the groups in terms of age or baseline IOP. Both groups included patients with mild, moderate, and advanced glaucoma, according to Hodapp–Parrish–Anderson. Initially, both groups showed a comparable baseline IOP and a similar number of localized antiglaucomatous drops. Baseline characteristics are shown in Table 1.

Both procedures resulted in significant IOP reduction postoperatively (iStent inject^®^ W group *p* = 0.032; Hydrus group *p* < 0.001, respectively). A comparison of the IOP reduction between the two groups showed a significantly greater reduction in the Hydrus group (*p* = 0.043; see Table 1 and Figure 2).

Furthermore, the Hydrus group demonstrated a significant reduction in the daily use of antiglaucoma eye drops (*p* = 0.002), while the iStent inject^®^ W group showed no significant reduction in medication use (*p* = 0.054, see Figure 2A). A significant difference was also found in the reduction in local therapy between both groups (*p* = 0.015, see Figure 2D).

The overall success rate was not significantly higher in the Hydrus group (38%) compared to the iStent inject^®^ W group (13%) (*p* = 0.102, see Table 1). We were able to demonstrate an MCID rate of ≥50% in both groups, which did not differ significantly. Both groups showed no complications according to the ESG criteria.

## 4. Discussion

MIGS is designed to lower IOP in patients with mild to moderate glaucoma while minimizing the risks associated with more invasive procedures, such as trabeculectomy (TE). Several studies have demonstrated that MIGS implantation in these patients could be a viable alternative in terms of surrogate success rates and long-term IOP control [7,15].

Among the various MIGS devices available in the European Union, the iStent inject^®^ W is one of the most widely used implants. It has demonstrated favorable outcomes in reducing IOP in patients with mild to moderate glaucoma in numerous studies [6,7].

In a study by Hengerer et al., the effectiveness of a double iStent inject^®^ (2nd generation) implantation without concurrent cataract surgery was evaluated in patients with POAG, PEX glaucoma, narrow-angle glaucoma, and secondary glaucoma [6]. This study reported an average IOP reduction of 42% and a significant reduction in the need for topical antiglaucoma medications. The average visual field defect in the study was a mean deviation (MD) of 6.4 dB, corresponding to grade II glaucoma according to the Hodapp–Parrish–Anderson classification [16].

While these findings suggest a significant improvement in IOP control, differences between the studies limit direct comparisons. Hengerer et al.’s patient cohort had a more advanced visual field defect preoperatively, and their study included patients who had undergone prior ocular surgeries, which were excluded from our analysis. Additionally, the lens status of the patients in Hengerer et al.’s study was not reported, limiting our ability to compare outcomes with our study, which included combined cataract surgery. Furthermore, it is unclear whether a washout period was implemented before surgery, which could affect baseline IOP measurements.

The IOP reduction observed in Hengerer et al.’s study appears to be more pronounced compared to our findings. However, previous studies have shown that IOP reduction from MIGS tends to decrease over time, and the 36-month results reported by Hengerer et al. may reflect this trend. Their study also showed a greater reduction in medication burden, with 82% of patients requiring fewer topical medications, compared to an average reduction in our study. One explanation for this could be the higher preoperative medication burden in Hengerer et al.’s cohort, with 75% of patients using three or more medications, compared to an average of 2.8 medications in our study. It is well established that patients with higher preoperative IOP and medication use tend to experience greater reductions in both IOP and medication burden following MIGS [17].

While our study did not allow for a direct comparison due to the limited number of patients with PEX glaucoma, these findings suggested that MIGS procedures, such as the iStent inject, can effectively lower IOP not only in POAG patients but also in those with PEX glaucoma. This highlights the broader applicability of MIGS for managing IOP across different types of glaucoma. Overall, the findings of Hengerer et al. are consistent with our study, though the differences in patient selection and methodology should be considered when interpreting the results.

The Horizon Study, which led to the approval of the Hydrus implant, was re-evaluated by Ahmed et al. in 2022 to assess the 5-year outcomes [18]. This study compared the efficacy of Hydrus implantation combined with cataract surgery to cataract surgery alone in patients with POAG. Over the 5-year period, the Hydrus group achieved an absolute IOP reduction of 8.3 mmHg, corresponding to a relative reduction of approximately 30%. Notably, 66% of patients in the Hydrus group were free of antiglaucoma medications at the 5-year mark, whereas only 43% of patients in our study were medication-free at 12 months.

It is important to recognize that the study designs differ. In the Horizon Study, Ahmed et al. defined baseline IOP without the use of antiglaucoma medications, resulting in higher preoperative IOP values and, consequently, a greater IOP reduction postoperatively. In contrast, in our study, we were unable to assess baseline IOP without medication, but there is evidence suggesting that the preoperative IOP in our cohort may have been lower due to ongoing therapy.

Despite these differences, the results of our study align with the 5-year findings of Ahmed et al. Both studies demonstrated a significant reduction in IOP at 12 months. However, the Horizon Study reported a greater reduction in the need for topical therapy, although 3.3% of patients required additional surgical interventions, such as selective laser trabeculoplasty.

As early as 2012, a review found that the IOP-lowering effect of cataract surgery alone is lower than the combination of cataract surgery and MIGS implantation [19]. This effect was confirmed in several more recent studies. One study by Lee et al. from 2020, for example, showed a significantly better IOP reduction through the implantation of iStent and Hydrus in combination with cataract surgery than lens surgery alone [20]. A 2023 systematic review analyzed the effects of various MIGS implants, with most studies focusing on device implantation without concurrent cataract surgery. The analysis of the iStent inject^®^ W included data from 41 individual studies, reporting IOP reductions of 11% to 30% in prospective studies and a 71 to 72% decrease in antiglaucoma medication use. These findings are consistent with our results regarding IOP reduction. In our study, we observed an average reduction of 9% in IOP 12 months after combined cataract surgery and iStent inject^®^ W implantation, despite the absence of a preoperative washout phase. However, we did not find a statistically significant reduction in antiglaucoma medication use.

For the Hydrus implant, six studies were evaluated, reporting an average IOP reduction of 26% in prospective studies, along with a 61% reduction in antiglaucoma medications. These findings are in line with our study, where we demonstrated a significant reduction in IOP with the Hydrus implant when combined with cataract surgery. In fact, our study showed a greater IOP reduction in the Hydrus group compared to the iStent inject^®^ W group.

The Horizon Study, which provided initial data on the Hydrus implant, defined complications or adverse effects such as peripheral anterior synechiae or device removal. Notably, we did not observe any complications in either of our study groups. It is worth mentioning that the Horizon Study represented the first clinical use of the Hydrus implant, and the surgeons involved were not yet fully trained with the device. In contrast, our study was conducted with specialized surgeons who had received training prior to performing the procedures. This higher level of surgical expertise may have contributed to the lower complication rates observed in our study [21].

A prospective study by Ahmed et al. compared the outcomes of implanting two iStent inject^®^ W devices with the standalone implantation of a single Hydrus microstent in patients with POAG [11]. The study analyzed approximately 150 eyes across nine countries, assessing both the IOPreduction and the decrease in medication use 12 months postoperatively. In both measures, the Hydrus Microstent demonstrated superior efficacy compared to the iStent inject^®^ W. These findings align with the results of our study, where we also observed a greater IOP reduction following Hydrus implantation compared to iStent inject^®^ W implantation.

It is unclear whether Ahmed et al. used the first or second generation of the iStent, as the study included patients from 2013 to 2015, suggesting that the first-generation implant may have been used. Additionally, the study did not involve combined cataract surgery with MIGS implantation, unlike our study, which evaluated outcomes in a combined surgical setting.

Ahmed et al.’s study focused exclusively on POAG patients, whereas in clinical practice, MIGS is sometimes used in patients with secondary glaucomas, such as PEXglaucoma or pigment dispersion syndrome. These conditions are associated with altered anterior chamber angles, increased pigmentation, and impaired aqueous outflow [22,23]. As a result, the implantation of devices like the Hydrus or iStent may be more challenging in these patients, and postoperative outcomes could differ from those seen in POAG patients.

Several studies have included not only patients with POAG but also those with PEX glaucoma. Hengerer et al. conducted a prospective cohort study demonstrating a significant reduction in IOP in patients with PEX glaucoma following combined cataract surgery and iStent injection implantation. In a subgroup analysis, the study reported a 32% reduction in IOP and a 64% decrease in the need for antiglaucoma medications in the PEX glaucoma group.

While our study does not allow for a direct comparison due to the limited number of patients with PEX glaucoma, these findings suggest that MIGS procedures, such as the iStent injection, can effectively lower IOP not only in POAG patients but also in those with PEX glaucoma. This highlights the broader applicability of MIGS for managing IOP across different types of glaucoma.

It remains unclear whether patients with advanced glaucoma (Hodapp stage III) derive significant benefit from MIGS procedures. Several studies published in 2024 have addressed this issue, specifically evaluating the outcomes of combined phacoemulsification and Hydrus implantation. Ang et al. reported a median reduction in two glaucoma medications 12 months postoperatively in patients with normal tension glaucoma and Hodapp III, although no significant reduction in IOP was observed [24].

In contrast, Oberfeld et al. examined the impact of combined cataract surgery with MIGS, including iStent and Hydrus microstents, in patients with severe glaucoma (Hodapp III). They observed a reduction in mean IOP from 16.7 mmHg preoperatively to 13.5 mmHg at 12 months, representing a 19% decrease [25]. Additionally, the study provided evidence of a reduction in the need for local therapy following surgery. These findings suggest that MIGS may offer some benefit in terms of IOP and medication reduction in patients with advanced glaucoma, though the outcomes may vary based on individual patient factors and the specific type of MIGS procedure performed.

All studies discussed above show a good IOP reduction with iStent and Hydrus implantation with POAG in mild to moderate glaucoma. There is also evidence that MIGS, in general, but especially Hydrus implantation, is also able to lower IOP in patients with PEX glaucoma and severe glaucoma (Hodapp III).

Nevertheless, this study has some limitations: due to the limited number of patients, no subgroup analysis regarding PEX and pigment dispersion glaucoma is possible. Furthermore, the real-life cohorts differ from each other. We included two patients with normal tension glaucoma in the iStent group but no patients in the Hydrus group. Also, the different types of glaucoma in both groups do not match perfectly. Therefore, more patients and bigger data are necessary to analyze outcomes in subgroup analysis.

Our study does not show any significant difference in the success rate defined by the EGS criteria. Due to low baseline IOP, the relative IOP reduction was sometimes less than 25% but showed good absolute reduction. The caveat of long-term efficacy still applies to both implants, as both the iStent Inject W and the Hydrus have only been available on the international markets for a few years. Future studies will show whether long-term complications such as dislocations can occur.

## 5. Conclusions

The findings of this study demonstrated a greater reduction in IOP following Hydrus implantation compared to iStent inject^®^ W implantation. The Hydrus implant occupies approximately 25% of the Schlemm’s canal, while the iStent inject^®^ W involves the placement of 1–2 stents in close proximity to each other. Our study showed a higher overall success rate and greater reduction in the need for antiglaucoma medications in the Hydrus group compared to the iStent group.

Overall, the Hydrus implant appears to be a safe and reliable MIGS option, providing slightly greater IOP reduction than the iStent inject^®^ W in real-world clinical settings. Additionally, the Hydrus implant seems effective not only in patients with POAG but also in those with PEX glaucoma, offering promising outcomes in both groups. Further subgroup analyses, particularly for specific patient populations such as those with PEX glaucoma and advanced glaucoma, are necessary to fully evaluate the efficacy of these devices across different clinical scenarios.

## Figures and Tables

**Figure 1 diagnostics-15-00493-f001:**
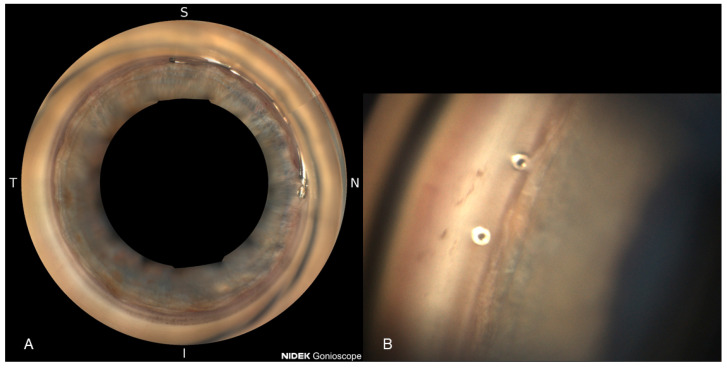
Hydrus^®^ Microstent (**A**) and iStent Inject W (**B**) visualized with NIDEK GS-1 Gonioscope (NIDEK, Gamagori, Aichi, Japan).

**Figure 2 diagnostics-15-00493-f002:**
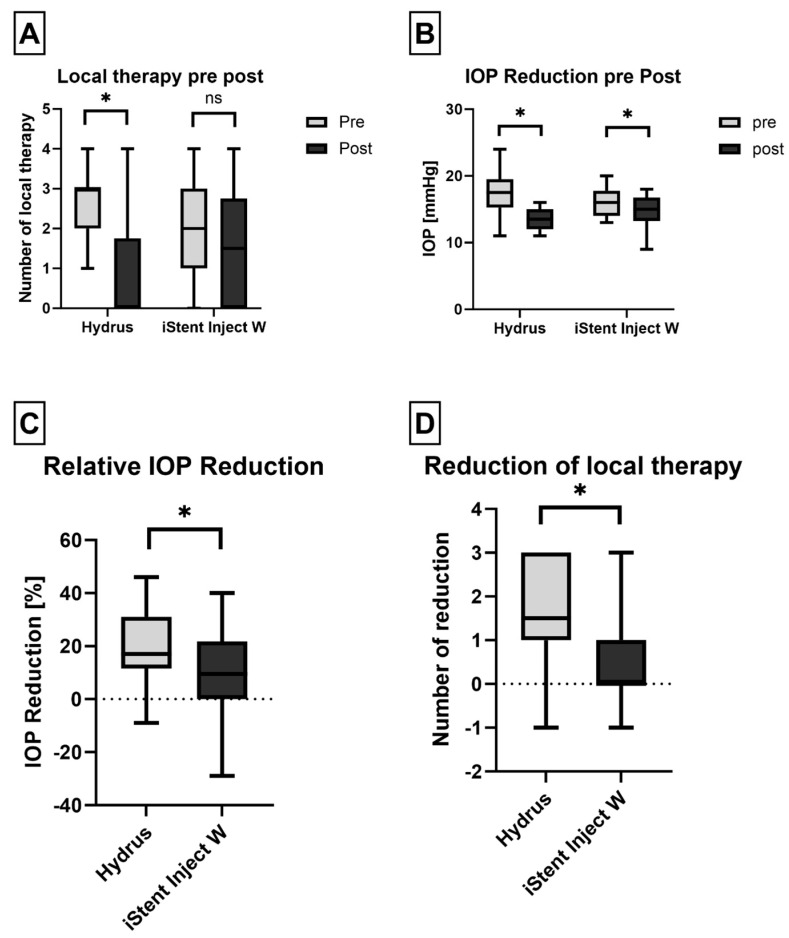
Both Hydrus and iStent Inject W showed a significant reduction in local therapy (**A**) and IOP (**B**). The relative IOP reduction was significantly higher in the Hydrus group (**C**). There was also a significant reduction in local therapy in the Hydrus Group compared to the iStent Inject W group (**D**); * *p* < 0.05.

**Table 1 diagnostics-15-00493-t001:** Demographics and baseline characteristics of the study cohort. Data on continuous variables are reported as the mean (±standard deviation).

	Hydrus	iStent Inject^®^ W	*p*-Value
Patients/eyes (n)	12/16	12/16	
Diagnosis			
- POA-Glaucoma	10	13	
- PEX-Glaucoma	6	1	
- Normal-Tension-Glaucoma	0	2	
Age (years)	71.9 ± 6.61	71.8 ± 4.6	0.985
Sex (m/f)	5/7	8/4	
Baseline Visual field MD (dB)	−10.3 ± 9.3	−3.4 ± 5.4	
Classification of glaucoma by Hoddap–Parrish–Anderson criteria			
I: MD 30-2: >−6.0	8	12	
II: MD 30-2: −6 to −12	3	3	
III: MD 30-2: <−12	5	1	
Baseline IOP (mmHg)	17.3 ± 3.4	16.0 ± 2.0	0.254
Postoperative IOP (mmHg)	13.4 ± 1.6	14.5 ± 2.3	
Relative IOP Reduction (%)	20.0 ± 13.5	9.0 ± 15.9	0.043
Number of antiglaucoma eye drops at baseline	2.7 ± 0.8	2.0 ± 1.1	0.080
Number of antiglaucoma eye drops after surgery	0.9 ± 1.3	1.5 ± 1.4	0.015
Success rate			0.102
Overall success	38%	12%	
Complete success	25%	6%	
Qualified success	13%	6%	
Failure	62%	78%	
MCID rate			
Absolute reduction (>2 mmHg)	81%	50%	0.288
Medication free (%)	56%	38%	0.063
Complications (hyphaema, hypotonia, device removal)	0	0	

POA = primary open angle; PEX = Pseudoexfoliative glaucoma; MCID = minimal clinically important difference.

## Data Availability

The data presented in this study are available on request from the corresponding author. The data are not publicly available due to privacy or ethical restrictions.

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
