# Peer review of "Comparison of the Intraocular Pressure-Lowering Effect of Minimally Invasive Glaucoma Surgery (MIGS) iStent Inject W and Hydrus—The 12-Month Real-Life Data†"

_diagnostics, 2025, doi:10.3390/diagnostics15040493_

Round 1

Reviewer 1 Report

Comments and Suggestions for Authors

- in the "Materials and Methods" section, there is NO information about the material. Please add.

- the "Results" section must be improved: tables and pictures are used to back-up the text, but in your manuscript, the whole "Results" section is composed of one big table, one picture, and very little text. Please improve this section.

- paragraph 118-121: you state that the iStent showed no significant reduction in medication use, then you state that "(...)A significant difference was also found in the reduction of local therapy between both groups (...)". These two statements are contradictory. Please modify.

- please add to your results a statistical comparison between the IOP lowering and the stage of the glaucoma (acoording to Hoddap-Parrish-Anderson criteria). I am asking for this because I noticed that you enroled patients from early, moderate, and advanced glaucoma stages, and it is known that IOP lowering postoperatively can vary depending on glaucoma stage.

- there is no control group in your analysis. Please add a control group (e.g. phaco+MIGS+/-topical medication vs phaco+topical medication) 

- Table 1: Overall success+Failure=101% (Hydrus) and Overall success+Failure=94% (iStent). Please modify.

- congratulations for 0 complications!

- lines 169-172 and lines 244-247 are the same; please delete one of them

Author Response

Review 1:

Comments and Suggestions for Authors

- in the "Materials and Methods" section, there is NO information about the material. Please add.

“Thank you for your comment. We used the iStent inject W device from Glaukos and the Hydrus Microstent device from Alcon, both were described in the introduction section We have added information about the materials used in this trial to the Methods section as per your request.”

- the "Results" section must be improved: tables and pictures are used to back-up the text, but in your manuscript, the whole "Results" section is composed of one big table, one picture, and very little text. Please improve this section.

“Thank you for your comment. We added more text describing the tables and figures (see line 116 to 118 and 130 to 132, as well as the minimal clinical important difference (MCID)rate Section (p value). Should you require further information, we are eager to provide those. ”

- paragraph 118-121: you state that the iStent showed no significant reduction in medication use, then you state that "(...)A significant difference was also found in the reduction of local therapy between both groups (...)". These two statements are contradictory. Please modify.

“Thank you for your comment. These are three different evaluations: 1st + 2nd: The reduction of local therapy before and after surgery for the iStent and the Hydrus group. 3. The comparison of the percentage reduction in local therapy between the two groups.

We found a significant reduction in the Hydrus group, but not in the iStent group. The percentage reduction between the groups was also significant. Thank you for raising your concerns regarding this section. In order to better communicate this point, we added the corresponding figure in line 122 and line 124”

- please add to your results a statistical comparison between the IOP lowering and the stage of the glaucoma (acoording to Hoddap-Parrish-Anderson criteria). I am asking for this because I noticed that you enroled patients from early, moderate, and advanced glaucoma stages, and it is known that IOP lowering postoperatively can vary depending on glaucoma stage.

“Thank you for your comment. We acknowledge that a subgroup analysis would be helpful to see the effects in each individual group. Unfortunately the n size in the mild and moderate groups is too little (n between 1 and 5).Therefore, we are unable to conduct a meaningful analysis wihtin these groups. We have added a section to the Discussion section of our manuscript to better communicate this issue(ll. 277 and 294)”.

- there is no control group in your analysis. Please add a control group (e.g. phaco+MIGS+/-topical medication vs phaco+topical medication) 

“Thanks for the advice. In this real life study we used clinical data for glaucoma patients which underwent iStent + phaco or Hydrus + phaco operation. There is a lot of evidence that both devices can lower IOP (see lines 46-59). And there is also evidence that phaco + MIGS is more efficient than phaco as standalone (PMID: 22978183.) In this study we wanted to see the performance in a real life setting between both devices, which is why we respectfully believe that the current study design does not require further adaptation.

Nonetheless, we strongly agree that further studies applying such designs are eagerly needed and therefore added  information in the discussion section to clarify (ll.   205-209)”

- Table 1: Overall success+Failure=101% (Hydrus) and Overall success+Failure=94% (iStent). Please modify.

“Thank you for the comment. There was a little rounding error. We corrected the mistake in Table 1”

- congratulations for 0 complications!

“Thank you!”

- lines 169-172 and lines 244-247 are the same; please delete one of them

“Thank you for your comment. We deleted the statement in lines 169-172”

Reviewer 2 Report

Comments and Suggestions for Authors

This study retrospectively compared glaucoma patients who underwent either Hydrus® Microstent or iStent inject® W implantation combined with cataract surgery in a single center. There are some questions should be clarified.

1.          The study included patients diagnosed 77 with glaucoma. Which type of glaucoma? Same for all patients?

2.          In table 1: Patients/eyes (n) 16/12à 16 patients 12 eyes?

3.          How to determine to use  Hydrus® Microstent or iStent inject®

4.          Same surgeon for all patients?

Author Response

Review 2:

Comments and Suggestions for Authors

This study retrospectively compared glaucoma patients who underwent either Hydrus® Microstent or iStent inject® W implantation combined with cataract surgery in a single center. There are some questions should be clarified.

  1. The study included patients diagnosed 77 with glaucoma. Which type of glaucoma? Same for all patients?

“Thank you for your comment. We included 16 patients in both groups, 32 patients in total. The types of glaucoma are shown in table 1. Both groups included POA-, and PEX Glaucoma. Additionally 2 Normal-tension Glaucoma patients were included in the iStent group”

  1. In table 1: Patients/eyes (n) 16/12à 16 patients 12 eyes?

“Thank you for your comment. We used 32 eyes in total from 24 patients. You are right, we mixed up the patients and eye sections. We have corrected this mistake in accordance with your comment”

  1. How to determine to use  Hydrus® Microstent or iStent inject®

“Thank you for your question. The procedure was determined by the attending senior surgeon. We added further explanations on how the surgery technique was decided on in lines 80-81 to increase transparency”

  1. Same surgeon for all patients?

“Thank you for your question. We had three different experienced and certified surgeons performing the surgery. We added this information to the manuscript in lines 81-82”

Reviewer 3 Report

Comments and Suggestions for Authors

This retrospective study of two devices used for controlling IOP in open angle glaucoma is sound but has not brought out any new data which can add to literature. The type of patients , especially secondary open angle glaucoma are not the same in the two arms of the study and therefore results will be affected .The sample size is small and so is the follow up of only one year where as studies , as quoted by the authors, have had follow up even upto 5 years! Do these devices really continue to keep IOP controlled after 5 years is something we all need to know. The device cost, surgery cost and post operative glaucoma medicines for so many years , though at  reduced due to decrease in number of medications, is prohibitive for an average patient .

Though every bit of scientific data adds to our knowledge especially when new devices are used, the study only adds some more data about the two devices which is already known and published.

Author Response

Review 3:

Comments and Suggestions for Authors

This retrospective study of two devices used for controlling IOP in open angle glaucoma is sound but has not brought out any new data which can add to literature. The type of patients , especially secondary open angle glaucoma are not the same in the two arms of the study and therefore results will be affected .The sample size is small and so is the follow up of only one year where as studies , as quoted by the authors, have had follow up even upto 5 years! Do these devices really continue to keep IOP controlled after 5 years is something we all need to know. The device cost, surgery cost and post operative glaucoma medicines for so many years , though at  reduced due to decrease in number of medications, is prohibitive for an average patient .

Though every bit of scientific data adds to our knowledge especially when new devices are used, the study only adds some more data about the two devices which is already known and published.

“Thank you for the comment!”

Round 2

Reviewer 1 Report

Comments and Suggestions for Authors

There is no material in your ‘Material and Methods’ section means that you have no data about the material your experiment is conducted on; this material is represented by the patients, not the MIGS devices - these are the methods you work with. Please add some information (eg. : number, stage, …)

Nothing else. 

Author Response

Dear Reviewers,

Thank you very much again for reviewing and giving us the opportunity to revise our manuscript. We thank all reviewer for their very helpful comments, which improved our manuscript significantly. The new manuscript was revised in accordance to the reviewers’ suggestions. Please see below for detailed answers. Changes in the “Manuscript with tracking changes”-file are highlighted in yellow. If any questions arise regarding the manuscript, please do not hesitate to contact us. We would like to thank you and the reviewers again for investing their time and appreciate getting the opportunity to submit our revised manuscript to your highly esteemed journal.

Sincerely,

Cedric Weich

Reviewer 2 Report

Comments and Suggestions for Authors

Thanks for your response. There are two cases of NTG in iStent group but none in Hydrus. It may affect the comparison of outcome, especially the relative IOP reduction in both groups. Different type of glaucoma in both group should list in the limitation.

Author Response

(The authors gave the same response as above.)
